# OpenReview forum: "JailDAM: Jailbreak Detection with Adaptive Memory for Vision-Language Model"
_colmweb.org/COLM/2025/Conference — COLM 2025_

### Official Review · Reviewer_mPCF · 2025-04-23

**Rating:** 6
**Confidence:** 2
**Ethics Flag:** 1

**Summary:**

The paper introduces JAILDAM, a black-box-compatible framework for detecting jailbreak attacks in vision-language models without requiring access to harmful training data or model internals. Instead of relying on gradients or fine-tuning, JAILDAM constructs a policy-guided memory bank of unsafe concepts (e.g., hate speech, privacy violations) generated by GPT-4o based on safety policies. At test time, it encodes multimodal inputs using CLIP, computes their attention alignment with the unsafe memory, and uses an autoencoder to detect anomalies via reconstruction error—flagging high-error inputs as potential jailbreaks. To adapt to some advanced attacks, JAILDAM includes a test-time adaptation mechanism that updates the memory with residual features from suspicious inputs. It also extends to JAILDAM-D, an end-to-end defense system that prepends refusal prompts to harmful queries. Experiments on three jailbreak benchmarks (MM-SafetyBench, FigStep, JailBreakV-28K) show that JAILDAM outperforms prior baselines in accuracy and efficiency, offering a practical and scalable solution to VLM safety.

**Reasons To Accept:**

1.without accessing to the internalmodel states or gradients, this makes the practical, and also applicable to closed-source VLMs.

2.The framework introduces a policy-guided memory bank of unsafe concepts, offering a fresh perspective on jailbreak detection using autoencoder-based method.

3.The paper claims the proposed method is 60x faster at inference than uncertainty-based mthods, and requires limited training. It makes a practical method.

**Reasons To Reject:**

1.The method requires CLIP-based embedding, and it might not be applicable for some API-only models.

---

> ### Author Response · Authors · 2025-05-31
>
> Thank you for acknowledging our contributions. In response to your concerns and potential weaknesses,first, we acknowledge that our model relies on CLIP-based encoders since it doesn’t require task-specific fine-tuning and highly efficient in vision-language tasks. While we use CLIP to encode image-text inputs, our detector can still be applied to both open-weight and black-box (API-only) VLMs, such as GPT-4o-mini (see Figure 4). This design allows our method to generalize across a wide range of VLMs without modification.
>
> On the other hand, our model do have some advantages compared with black-box defense methods such as the Adashield. Adashield-S adopts a single static prompt, which results in relatively lower performance across various models, including Llava-13B, CogVLM-chat-1.1v, and GPT-4o-mini (see Table 5). While Adashield-A employs a dynamic prompting strategy that requires harmful training data for iterative adjustment and incurs significantly higher inference & training time cost. In contrast, our method requires no access to harmful data, achieves stronger or comparable performance across models, and maintains much faster inference & training efficiency.
>
> Overall, our method strikes a balance between generalizability, efficiency, and robustness, making it a practical solution for real-world jailbreak detection.

---

> > ### Comment · Reviewer_mPCF · 2025-06-11
> >
> > I appreciate the author's detailed rebuttal. However, my concern remains that reliance on CLIP embeddings inherently limits applicability to other VLM models that do not expose embeddings or intermediate representations, potentially constraining real-world deployment. Therefore, I maintain my original score.

---

### Official Review · Reviewer_gGas · 2025-05-11

**Rating:** 7
**Confidence:** 5
**Ethics Flag:** 1

**Summary:**

This paper presents a well-structured and highly relevant study on jailbreak detection for vision-language models. The proposed JAILDAM framework effectively addresses the key challenges in existing detection methods, offering innovative solutions in terms of black-box compatibility, computational efficiency, and data-free training. The experimental evaluation is comprehensive, with solid results demonstrating its superiority over baselines. However, the reliance on GPT-4o for memory bank generation raises concerns about cost and scalability, and the framework's performance in extremely complex real-world attack scenarios remains to be fully explored.

**Reasons To Accept:**

1.Significant Problem-Solving
The paper addresses the critical challenge of detecting jailbreak attacks in multimodal large language models, a growing concern in real-world applications. JAILDAM effectively mitigates the limitations of prior work, such as dependence on hidden states and fully labeled harmful datasets.

2.Novel Methodology
JAILDAM introduces a novel policy-driven memory mechanism combined with an autoencoder-based detection pipeline. The use of GPT-4o to build a dynamic memory bank of unsafe concepts is both creative and practical for black-box settings.

3.Comprehensive Experiments
The experiments are thorough, covering multiple VLM jailbreak benchmarks and strong baselines. Metrics such as AUROC and F1-score demonstrate JAILDAM’s clear superiority in both accuracy and efficiency.

**Reasons To Reject:**

1. Methodological Dependence
The reliance on GPT-4o to generate the memory bank of unsafe concepts poses a significant concern. This approach may limit the reproducibility and generalizability of the method. Since GPT-4o is a proprietary model, its availability, cost, and potential changes in behavior over time could affect the performance of JAILDAM. Additionally, using a large-scale language model for this task might introduce biases present in GPT-4o, leading to inaccurate detection of jailbreak attempts.

2. Insufficient Robustness Analysis
Although the experiments demonstrate good performance on existing benchmarks, the evaluation of JAILDAM's robustness against emerging and complex jailbreak techniques is lacking. As jailbreak strategies are constantly evolving, the model's ability to handle highly sophisticated or novel attacks remains uncertain. Without comprehensive stress-testing under various real-world-like adversarial scenarios, it is difficult to ensure the long-term effectiveness of the proposed framework.

---

> ### Author Response · Authors · 2025-05-31
>
> Thanks for recognizing our contributions in this work. For your concerns and potential weakness, we have the following response:
>
> ### **Methodological Dependence on GPT-4o**
>
> - (1) **Memory Bank Generation**:
>  While we rely on GPT-4o to generate the unsafe memory bank in our implementation, our method is not tied to this specific model. Instead, it provides a generalizable strategy for leveraging LLMs to produce diverse harmful concepts, serving as a scalable alternative to manual curation. To validate the generality of our approach, we additionally generate memory banks using *DeepSeek-R1* under the same prompt and do the same experiments. As shown in the performance table below, detectors built on these alternative concept sources yield results that are consistent with our main findings, demonstrating similarly strong performance to the GPT-4o-based setup. **This shows that JailDAM is a flexible framework that we can leverage different LLMs.**
>
> | **Model**         | **MM-SafetyBench**  | **FigStep**        | **JailBreakV-28K**   |
> |-------------------|---------------------|---------------------|-----------------------|
> | Original (GPT-4o) | 0.9472 / 0.9155     | 0.9608 / 0.9616     | 0.9465 / 0.9464       |
> | DeepSeek-R1       | 0.9385 / 0.9253     | 0.9231 / 0.9236     | 0.9238 / 0.9300       |
>
> *Table 1: Detection Performance (AUROC / AUPRC) across Different Memory Generators*
>
> - (2) **Bias of using GPT-4**:  You raise an important point: relying solely on a single large-scale language model, such as GPT-4o, for harmful memory generation (as in JailDAM) may introduce biases inherent to that model. In our case, we adopted GPT-generated memory primarily due to the lack of a well-defined, standardized harmful content policy. However, this design choice can adapt to real-world settings where different organizations typically enforce their own tailored policies, which helps mitigate the issue of model bias by grounding detection in concrete, context-aware rules rather than relying on a single, potentially biased model’s judgment.
>
> ### **Insufficient Robustness Analysis**
> Thank you for your valuable feedback. Your suggestion offers a meaningful direction for improving JailDAM. As clarified in the introduction, our work specifically focuses on structure-based jailbreak attacks, which differ from traditional adversarial attacks in both intent and mechanism. While adversarial attacks are indeed an important area of research, they fall outside the current scope of this paper. That said, we agree that exploring both adversarial and structure-based methods could be a promising direction for future work.  We will clearly acknowledge this limitation and future direction in the revised paper by adding the following section:
>
> ```
> Limitation and Future Work
>
> This study focuses specifically on structure-based jailbreak attacks and does not encompass adversarial attacks such as perturbation-based or gradient-based methods. While this focused approach allows for in-depth analysis of structure-based vulnerabilities, it represents a limitation in the comprehensiveness of our detection framework. Future research will extend this work by developing more generalizable detection mechanisms capable of identifying both structure-based and adversarial attacks within a unified framework.
> ```

---

> > ### Comment · Reviewer_gGas · 2025-06-04
> >
> > Thank you to the authors for their thorough and well-reasoned response to the review comments, effectively addressing my main concerns with detailed explanations and additional experiments.

---

### Official Review · Reviewer_SCGg · 2025-05-13

**Rating:** 6
**Confidence:** 3
**Ethics Flag:** 1

**Summary:**

Most of exiting methods are designed with some limitations: 1. while box setting that needs to access the hidden states; 2. need high computational overhead; 3. Depend on fully labeled harmful data. For handling these limitations, this paper focuses on designing a test/inference time adaptive framework JailDAM that utilizes a memory-based method by a policy-driven unsafe knowledge representation. This method will have a better generalization to unseen jailbreaking strategies while maintaining a reasonable efficiency. The key of this JailDAM is a autoencoder based detection solution that utilized the memory based attention feature encoding to detection jailbreaking attack.

**Questions To Authors:**

Please provide more explanation about the generalization capability of proposed method.

**Reasons To Accept:**

- This paper proposes a jailbreak detection framework that can be applied to black-box models especially these proprietary VLMs.

- The proposed framework JailDAM achieved a very reasonable performance without the need of expensive data collection and perturbation.

- A memory based system can update the unsafe knowledge at inference time. Due to this test-time adaption, it improves the robustness of handling new jailbreaking types or strategies.

**Reasons To Reject:**

- This paper mentioned formulation 4 can ensure the detector can be trained using safe data and policy constraints. It minimizes the relation between safe data with unsafe constraints. It highly depends on the safe data. I am still worried about the claim about the good generalization capability. For instance, I have a safe data that is related to “travel” topic. The policy is about Privacy violence in health domain. If the input is from another domain like insurance with totally difference data distribution, it is challenging to detect the relationship between unseen data with the policy. More evidences need to be provided or analyzed to prove it can be generalized to other domains.

---

> ### Author Response · Authors · 2025-05-31
>
> We sincerely thank you for the insightful comments and constructive feedback. We agree that the generalization ability of a detector trained solely on safe data and policy constraints can be influenced by the coverage and quality of the conceptual memory bank. To address this concern, we provide a two-part response supported by empirical evidence:
>
> ---
>
> **1. Memory Comprehensiveness**
> Our first point is that the detector’s ability to generalize across domains benefits significantly from having a comprehensive memory bank. Even when the training and testing domains differ, the model can still capture harmful information because harmful policies are well-covered in the memory bank.
>
> Moreover, even when the memory bank is incomplete, we emphasize that our method is equipped with an **Adaptive Memory Mechanism**, which mitigates the limitations of incomplete memory coverage. To verify this, we conduct a memory removal experiment on MMSafetyBench dataset. We start with a full memory bank built from the 13 harmful categories defined in MMSafetyBench. We then progressively remove 1, 3, 5, and 7 categories and evaluate the model’s performance. As shown below, the detection accuracy degrades slowly as the memory coverage becomes sparser. This demonstrates that our memory mechanism is robust even when the memory of unsafe policy is not complete.
>
> | **Metric** | **Remove 0 Class** | **Remove 1 Class** | **Remove 3 Class** | **Remove 5 Class** | **Remove 7 Class** |
> | ---------- | ------------------ | ------------------ | ------------------ | ------------------ | ------------------ |
> | **AUROC**  | 0.9472             | 0.9270             | 0.9140             | 0.9020             | 0.8350             |
> | **AUPR**   | 0.9155             | 0.9046             | 0.9068             | 0.8785             | 0.7917             |
>
> **Table 1: Performance of JailDAM After Removing Memories**
>
>
> **2. Out of Domain Generalization to Benign Inputs**
> To directly assess whether the detector generalizes to the out-of-domain test input, we conduct a second experiment using *MMMU* [1] as a different domain test input. Unlike *MM-vet* focusing on general VQA, *MMMU* consists of academic-style content. We compared the difference of 2 datasets as: $$ \mathbb{E}[\cos(\mathbf{q}_i, \mathbf{q}_j)] \approx 0.11
> $$ where $\mathbf{q}_i, \mathbf{q}_j$ be textual question embeddings. This indicate a substantial semantic shift between *MM-vet* and *MMMU*. We then apply the detector (trained solely on *MM-vet*) to *MMMU* and evaluate whether it can correctly classify the inputs as safe. The results are summarized below:
>
> |**Benign Dataset**| **Jailbreak Dataset** | **AUROC** | **AUPR**  |
> | ---------------------- | --------- | --------- | --------- |
> |MMMU  | Jailbreak-28k     | 0.9034 | 0.8962 |
> |MMMU   | MMSafetyBench    | 0.9452 | 0.9396 |
> |MMMU  | Fig-step     | 0.8852 | 0.8766 |
>
> **Table 2: Out of Domain Generalization of JailDAM**
>
> The detector achieves strong performance, demonstrating that our model's robustness and captures **generalizable, policy-aligned safety signals**.
>
> Together, JailDAM has comprehensive memory bank with adaptive memory, OOD Generalization to Benign Inputs indicating that our detector exhibits **strong domain generalization** and does not simply memorize training distribution artifacts.
>
> ---
> [1] Yue, Xiang, et al. "Mmmu: A massive multi-discipline multimodal understanding and reasoning benchmark for expert agi." Proceedings of the IEEE/CVF Conference on Computer Vision and Pattern Recognition. 2024.

---

### Author Response · Authors · 2025-06-07

Dear reviewers,

Thank you again for your valuable time and thought-through comments. We have provided thorough responses and additional results.

While we are encouraged to find one reviewer raised the score, we would appreciate it if other reviewers could read our responses and see if the concerns have been addressed. Thank you.

Best regards,

Authors

---

### Decision · Program_Chairs · 2025-07-08

**Decision:**

Accept

**Comment:**

JailDAM is a novel framework that detects "jailbreak" attacks in multimodal large language models (MLLMs). It overcomes limitations of existing methods—reliance on internal model access, high computational cost, and need for labelled harmful data—by using a memory-based approach guided by policy-driven unsafe knowledge. This allows dynamic adaptation to new attacks without explicit exposure to harmful content, demonstrating improved accuracy and speed in harmful content detection.

Reviewer SCGg highlights that one of the downsides of the proposed method is the generalisation power to unseen data distributions. The authors do acknowledge this as a weakness but provide an additional set of experiments that showcase how this method is able to mitigate this issue. There is definitely a drop in performance when changing the benign dataset (e.g., dropping certain categories). This set of experiments would be a useful addition to the paper camera-ready version.

Reviewer gGas highlights how this method entirely relies on GPT4o for memory generation which could be a problem for reproducibility.  As part of the rebuttal, the authors also investigate the use of DeepSeek-R1 for the memory generation component. I believe this additional test makes the paper even more robust. Performance with DeepSeek-R1 is almost on par with GPT4o, showcasing that it’s possible to get reasonable performance using this framework with open-weight models as well. The same reviewer raises another point regarding bias in the used language model; this is an important consideration that the authors acknowledge in the rebuttal, but that should be specifically discussed in the "limitations" section of the paper.

Reviewer mPCF mentions that one of the issues of this paper is the use of the CLIP embeddings because the method cannot be applied to closed-source models. However, this is a weakness that is incorrectly specified, considering that the method *can be applied* to any image/text input pair. The authors clarify this aspect in their rebuttal. Specifically: “While we use CLIP to encode image-text inputs, our detector can still be applied to both open-weight and black-box (API-only) VLMs, such as GPT-4o-mini (see Figure 4). This design allows our method to generalise across a wide range of VLMs without modification.”

Overall, the authors present a really interesting approach to mitigating VLM jailbreaks, and I believe this paper should be accepted.

[Automatically added comment] At least one review was discounted during the decision process due to quality]